# Analysis of Ventilation Efficiency as Simultaneous Control of Radon and Carbon Dioxide Levels in Indoor Air Applying Transient Modelling

**DOI:** 10.3390/ijerph19042125

**Published:** 2022-02-14

**Authors:** Mateja Dovjak, Ožbej Vene, Janja Vaupotič

**Affiliations:** 1Buildings and Constructional Complexes, Faculty of Civil and Geodetic Engineering, University of Ljubljana, 1000 Ljubljana, Slovenia; ozbejvene@gmail.com; 2Department of Environmental Sciences, Jožef Stefan Institute, 1000 Ljubljana, Slovenia; janja.vaupotic@ijs.si

**Keywords:** ventilation, residential buildings, transient modelling, radon, carbon dioxide

## Abstract

The impact of ventilation efficiency on radon (^222^Rn) and carbon dioxide (CO_2_) concentrations in the indoor air of a residential building was studied by applying transient data analysis within the CONTAM 3.4 program. Continuous measurements of ^222^Rn and CO_2_ concentrations, together with basic meteorological parameters, were carried out in an apartment (floor area about 27 m^2^) located in Ljubljana, Slovenia. Throughout the experiment (October 3–15), frequent ventilation (several times per day), poor ventilation (once to twice per day) and no ventilation scenarios were applied, and the exact ventilation and occupancy schedule were recorded. Based on the measurements, a transient simulation of ^222^Rn and CO_2_ concentrations was performed for six sets of scenarios, where the design ventilation rate (DVR) varied based on the ventilation requirements and recommendations. On the days of frequent ventilation, a moderate correlation between the measured and simulated concentrations (*r* = 0.62 for ^222^Rn, *r* = 0.55 for CO_2_) was found. The results of the simulation indicated the following optimal DVRs: (i) 36.6 m^3^ h^−1^ (0.5 air changes per hour, ACH) to ensure a CO_2_ concentration below 1000 ppm and a ^222^Rn concentration below 100 Bq m^−3^; and (ii) 46.9 m^3^ h^−1^ (0.7 ACH) to ensure a CO_2_ concentration below 800 ppm. These levels are the most compatible with the 5C_Cat I (category I of indoor environmental quality, defined by EN 16798-1:2019) scenario, which resulted in concentrations of 656 ± 121 ppm for CO_2_ and 57 ± 13 Bq m^−3^ for ^222^Rn. The approach presented is applicable to various types of residential buildings with high overcrowding rates, where a sufficient amount of air volume to achieve category I indoor environmental quality has to be provided. Lower CO_2_ and ^222^Rn concentrations indoors minimise health risk, which is especially important for protecting sensitive and fragile occupants.

## 1. Introduction

A built environment is defined as a four-dimensional human-made space that ranges from indoor to outdoor and provides the setting for human activity [1,2]. As a primary health determinant, it accounts for almost 20% of all deaths in the WHO European Region that are attributable to a degraded urban environment and housing-related inequalities, among which poor air quality presents a major contribution [3]. However, indoor air is often more seriously polluted than outdoor air, even in the largest and most industrialised cities [4]. It may contain over 900 chemicals, including particles and biological materials with potential health effects [5].

The building is an individual component of the built environment that contributes positively or negatively to both built and natural environments [6]. The design of buildings, either residential or non-residential, should follow the morphology of their engineering design [7], which will be defined and shaped by the context of the human-environment relationships [6]. The main interface between the indoor and outdoor environment is the building envelope (i.e., with transparent and nontransparent parts of the walls, floor and roof), which enables a continuous transfer of heat, mass and information through a medium (solid, fluid or gas) [7]. Human interventions in the built environment must be sustainable and not cause environmental degradation, to prevent negative impacts upon the occupants’ health and well-being. Among them, ventilation is essential to ensure the breathable air is healthy, by diluting pollutants originating in the building and removing them [8].

Two significant pollutants regulated by international and national legislation in the built environment are radon (^222^Rn) and carbon dioxide (CO_2_). Both can accumulate in the indoor air but are most often reduced rapidly with proper ventilation. The indoor/outdoor ratio (I/O) may vary for ^222^Rn from approximately 2 to over 100 [9], and for CO_2_ from 2 to 10 [10,11].

Radon (^222^Rn, Rn) is a radioactive noble gas that accumulates in the indoor air of insufficiently ventilated buildings and may increase lung cancer risk. It is ranked as the second most common cause of lung cancer immediately after smoking [9,12]. ^222^Rn is primarily formed by the α-transformation of radium (^226^Ra) in the earth’s crust, from where it migrates towards the surface via diffusion and advection and exhales in the atmosphere. In general, outdoor radon concentrations are low (about 10 Bq m^−3^) [13], depending on the geological characteristics of the terrain and the atmospheric mixing state [14,15]. On the other hand, indoor radon concentrations are usually higher by one order of magnitude (up to several 100 Bq m^−3^) or even more (up to several 1000 Bq m^−3^). There are four possible sources of radon entering a building: (1) the soil beneath the building, if the building envelope is leaky in contact with the ground; (2) construction products containing radium (e.g., fly-ash bricks); (3) tap water, insofar as it is obtained from groundwater sources, such as springs, wells and boreholes, which generally have higher radon levels than surface waters (rivers, lakes and reservoirs); and (4) natural gas released into the air via combustion. By far, the most important is the first source. In buildings with elevated indoor radon concentrations, only mitigation measures can adequately reduce radon entry into the building. Otherwise, if the radon concentration is close to or slightly exceeds the reference value, adequate and regular ventilation (natural, mechanical or hybrid) can significantly reduce the radon levels.

CO_2_ in indoor air is a metabolic product, a bio-effluent. It is the crucial indicator of room ventilation and a well-established measure of good indoor air quality (IAQ) [16,17]. Typical background CO_2_ concentration in outdoor ambient air is 350 to 450 ppm [18], where the dominant factor for the emissions is fuel combustion. The indoor concentrations of CO_2_ depend on the occupancy load, the room size, and the qualitative and quantitative ventilation characteristics. A range of 600 to 800 ppm of CO_2_ provides reliable indoor air quality, with an upper limit of 1000 ppm. Concentrations above 1000 ppm can lead to an increase in absenteeism, lower attendance, and reduced productivity. The maximum workplace concentration over 8 h is 5000 ppm, and the critical, only short-term, exposure concentration range is 6000 to 30,000 ppm. The effects of different CO_2_ concentrations [18] are an increased breathing frequency, headache (3–8%), nausea, vomiting, loss of consciousness (>10%), rapid loss of consciousness, and death (>20%).

Among the quantitative aspects of ventilation, the crucial parameter is the design ventilation rate (DVR), defined by legal requirements and/or recommendations [16,19,20,21,22,23,24,25]. The DVR can be determined as the amount of fresh air: (i) per floor area; (ii) per room volume; (iii) per occupant; and (iv) for a specific contaminant. The final selection of the DVR is, therefore, often left to the designer, who, intending to achieve the lowest possible ventilation heat losses, favours lower DVR values [26]. Persley [27] highlighted the problem that many practitioners and researchers claim a building has good IAQ because it complies with the 1000 ppm CO_2_ limit set in the standard. Therefore, identifying relevant CO_2_ concentrations that correspond to ventilation rate requirements must consider the building type and its occupancy, as well as other contaminants [27]. This aspect is essential, especially for the control and prevention of radon entry. For example, a Decree on the national radon programme [28] defines that ventilation is the primary measure required in all buildings where indoor radon concentrations are right below the 300 Bq m^−3^ (i.e., the reference level of the annual average indoor radon concentration in living and working spaces). In other buildings with radon concentrations above the reference level, it is necessary to set up an active radon ventilation system and seal the structural assemblies in contact with the ground to prevent radon entry.

The tightening of the building energy efficiency requirements, especially after 2010 [29,30], has been reflected in increased building airtightness as well as decreased DVRs in several ongoing construction projects [26]. As a result, such engineering measures might be related to a deterioration in indoor environmental quality. IAQ in energy-efficient residential and non-residential buildings has already been analysed by many authors [26,31,32,33,34,35,36,37,38,39]. The problem of increased indoor radon concentrations in renovated residential buildings has also been highlighted in several studies [32,35,36,37]. As reported, the concentration of ^222^Rn was increased from 17.5 to 49.6 Bq m^−3^ (11–33%) [36] and for 32 Bq m^−3^ (20%) [37] right after the building energy renovation. Similarly, the surveys of CO_2_ in new and renovated residential [39] and non-residential buildings [33,38] showed an increase in CO_2_ concentrations during occupancy to 2500 ppm [33,38] and 3000 ppm [39] (approximately 5–6 times, if the initial CO_2_ concentration is about 500 ppm). The increased CO_2_ concentrations were associated with lower ventilation rates, particularly in younger dwellings [40] that are naturally [40,41,42,43] or mechanically ventilated [43,44].

To provide an in-depth analysis of ventilation efficiency, some authors have included simulations of the selected indoor air pollutants in their studies and compared them to measurements. Concerning ^222^Rn, García-Tobar [45] proposed a methodology for estimating radon levels in a naturally and mechanically ventilated dwelling in a radon-prone area by using the CONTAM program. Further, García-Tobar [46] analysed the weather factors on indoor radon concentration in a new multistorey building in a radon-prone area. In the next study, García-Tobar [47] used CONTAM and computational fluid dynamics (CFD) transient simulations, including weather effects. Several authors have performed a transient simulation of CO_2_ concentrations in residential buildings and compared them to measurements. Szczepanik-Ścisło and Flaga-Maryańczyk [44] focused on a bedroom in a passive house. Using the CONTAM tool, the influence of occupancy schedules and the ventilation efficiency on the CO_2_ concentration was analysed over 10 days. According to the literature review, there has been an increased focus on the relationship between ^222^Rn or CO_2_ and ventilation efficiency. However, to characterise IAQ and the effectiveness of ventilation, it is crucial to identify the relevant ^222^Rn and CO_2_ levels simultaneously with those corresponding to ventilation rate requirements.

Our study focuses on the ventilation efficiency of a residential building. The primary purpose was to use a transient analysis of ^222^Rn and CO_2_ concentrations simultaneously for the first time using the CONTAM 3.4 program [48]. Methodologically, our research was divided into four steps. In the first step, a ventilation zone based on an actual apartment was modelled. In the second step, measurements of ^222^Rn and CO_2_ concentrations, together with basic meteorological parameters (air temperature, relative air humidity, barometric pressure), were conducted, and an accurate schedule of window opening was recorded. Based on the measurements, a model validation was carried out in the third step. In the fourth step, six sets of scenarios were critically analysed, defined by legal requirements and recommendations for the ventilation of residential buildings. Based on the findings, recommendations with practical benefits for constructions and renovations were developed, especially those where more efficient ventilation is sufficient as a radon protection measure.

## 2. Materials and Methods

### 2.1. Study Design

The study was conducted according to the steps below, which are explained in detail in the following sub-chapters.
Selecting the measurement location for indoor (an apartment) and outdoor measurements (meteorological and air quality station);Defining the ventilation zone in the apartment;Determining the schedule for the ventilation of the apartment;Conducting the measurements of ^222^Rn and CO_2_ concentrations and selected meteorological parameters (*T*–air temperature, *RH*–relative air humidity, *P*–barometric pressure);Simulating measured ^222^Rn and CO_2_ concentrations in the air of the apartment by using the CONTAM 3.4 [48] program;Validating the model;Verifying six ventilation scenarios for ^222^Rn and CO_2_ concentrations in the apartment.

### 2.2. Selection of Locations for Indoor and Outdoor Measurements

The study was conducted in Ljubljana (299 m above sea level, a.s.l.), the capital of Slovenia, located in the Ljubljana Basin in the central part of the county. It is characterised by a continental climate (Koppen–Geiger classification Cfb [49]) with an average minimum daily temperature of 5 °C and a maximum of 17 °C in October (time of measurements).

Two locations (one indoors and one outdoors) were selected for the measurements (at a distance of approximately 3 km from the city centre and approximately 2 km from each other):Indoor air measurements: A small apartment in an apartment building, part of a larger settlement in the city;Outdoor air measurements: The central meteorological and air quality station at the Environment Agency of Slovenia (ARSO).

### 2.3. Ventilation Zone

The ventilation zone was modelled according to the dimensions of an actual apartment in the apartment building. The building is a part of a larger settlement of apartment buildings and terraced houses built in 2002. In the basement, below the entire surface of the settlement, there is a garage with parking lots, which has a mechanical ventilation system installed. The apartment has a net size of 4.51 m × 6.33 m (26.6 m^2^ of net floor area, *A*_u_), with a height of 2.60 m (69.3 m^3^ of conditioned volume, *V*_e_). It faces east and is located on the 3rd floor (Figure 1) of a three-storey apartment building. The exterior wall assembly consists of reinforced concrete (16 cm) and facade plaster. The apartment is naturally ventilated by two French doors, with dimensions of 2.25 m × 2.70 m. Additional ventilation is possible through the kitchen hood and bathroom fan. Heating is based on a gas central heating boiler. The geometry of the ventilation zone with the position and dimensions of the openings is consistent with the actual apartment. The occupational load is 1.

### 2.4. Ventilation Schedule

The ventilation schedule of the apartment (with the day of the week, date, absence of occupant, and ventilation duration) is presented in Table 1. Throughout the experiment, only one door, the same French door, was open in full-screen mode (on the left side from the entrance). During the periods without ventilation, all of the French doors were closed, and the door to the bathroom was open (Figure 1). In addition, the kitchen hood and bathroom fan were not used during the measurement period.

In the first part of the measurement period, October 3–8, the schedule for the window opening (i.e., frequency and ventilation duration) was adjusted to maintain the CO_2_ concentrations below 1000 ppm. In the second part, October 9–10 (weekend), on Saturday, the dwelling was not ventilated and on Sunday, the previous ventilation regime was applied. In the last part, October 11–15, the ventilation was minimised to twice per day (Monday) and once per day (Tuesday and Thursday), with no ventilation on Wednesday.

### 2.5. Measurements

The measurements were conducted in the period 3–15 October 2021, and all presented data are reported in local time (LST = UTC + 2 h). A standardised protocol for characterising IAQ in residential buildings was followed [16,17,19,25,50]. In the apartment, the instrument for continuous measurement of the selected parameters was placed in the respiratory zone (living zone) at the height of 1.1 m above the floor; 3 m from the external window and wall, door and radiator; and 0.8 m from the internal wall (Figure 1) [16,17,19,25,50].

The selection of instruments was based on the expected radon (^222^Rn) concentrations in indoor and outdoor air and the requirements of our radon laboratory [51], accredited according to ISO/IEC 17025 [52]. Both devices were operated continuously in a diffusion mode with a frequency of once per hour.

Indoor air: radon *C*_Rn-in_ [Bq m^−3^] and carbon dioxide $C_{{\mathrm{CO}}_{2}}$ [ppm] concentrations, room air temperature *T*_in_ [°C] and relative air humidity *RH*_in_ [%] were measured with the Radon Scout Professional device (Sarad). The Radon Scout Professional monitor operates in the range from 0 Bq m^−3^ to 2 MBq m^−3^ with the sensitivity to Rn > 2.5 cpm/(kBq m^−3^). The integration interval of the data should be adjusted to the concentration range. If the expected radon concentrations are of the order of the reference level of 300 Bq m^−3^ or below, an interval of 60 min should be used. The sensor for CO_2_ operates in the range of 400 to 5000 ppm [53]. The integrated CO_2_ sensor uses the non-dispersive infrared (NDIR) operational principle.

Outdoor air: radon *C*_Rn-out_ [Bq m^−3^] concentration, temperature *T*_out_ [°C], relative humidity *RH*_out_ [%], and pressure *P*_out_ [hPa] were measured with the AlphaGUARD (Bertin Instruments) monitor, placed into a Stevenson screen at a height of 1.5 m above the ground. The instrument operates in the range from 2 Bq m^−3^ to 2 MBq m^−3^, and the efficiency of the detector is 1 cpm at 20 Bq m^−3^ [54].

### 2.6. Simulation

The simulation was based on the CONTAM 3.4 program [48]. This is a multizone analysis program, designed to analyse the IAQ in relation to the selected contaminants, ventilation rates, and the effectiveness of ventilation. According to the net dimensions of our test apartment, one ventilation zone was modelled. Openings (French doors, interior door) for natural ventilation were considered as airflow paths in our model.

Conservation of mass was applied to the zone, leading to a set of nonlinear algebraic equations that must be solved interactively. The detailed calculation protocol is presented in the CONTAM user guide [48]. The selected type for our analysis was transient and followed all of the required steps presented in the work by García-Tobar [45,46].

The input data in our model are as follows:i.Airflow paths: one-way flow using power law for French door and two-way flow for the indoor door (type of model); orifice area data for French door and one opening for the interior door (selected formula); 13,500 cm^2^ for French door and 20,000 cm^2^ for the interior door (cross-sectional data); 1.3111 cm for French door (hydraulic diameter); 30 for French door (Transition Reynolds number); 0.78 for French door and 0.78 for the interior door (discharge coefficient); 0.5 for French door and 0.5 for the interior door (flow exponent). The program enables a simultaneous mass balance of air in the ventilation zone to determine zonal pressures and airflow rates through each airflow path.ii.Measured data in outdoor air (hourly weather data, [55]): radon concentration *C*_Rn-out_ [Bq m^−3^], temperature *T*_out_ [°C], relative humidity *RH*_out_ [%], pressure *P*_out_ [hPa], and wind speed v*_w_* [m s^−1^].iii.Measured data in indoor air: radon concentration *C*_Rn-in_ [Bq m^−3^], carbon dioxide concentration $C_{{\mathrm{CO}}_{2}}$ [ppm], temperature *T*_in_ [°C], and relative humidity *RH*_in_ [%].iv.Default data: the radon generation rate [Bq h^−1^] was determined for every hour according to the methodology defined in Dovjak et al. [26]. The CO_2_ metabolic emission rate is 0.0027 dm^3^ s^−1^ during sleeping and 0.0038 dm^3^ s^−1^ during light activity [56]. The outdoor CO_2_ concentration is 400 ppm. Uncontrolled ventilation is 0.1 air changes per hour, ACH (6.9 m^3^ h^−1^).v.Defined schedules: the ventilation schedule of the apartment and the presence of the occupant were determined according to the records (Table 1).vi.Defined type of calculation: transient calculation of airflows and concentrations of ^222^Rn and CO_2_. The ^222^Rn and CO_2_ concentrations were determined from predefined indoor and outdoor sources. The main characteristics of ^222^Rn are an atomic weight of 222 kg kmol^−1^, a diffusion coefficient in the air of 5.91 mm^2^ s^−1^, and a half-life of its α-transformation of 3.8 days [45]. The main characteristics of CO_2_ are an atomic weight of 44 kg kmol^−1^ and a diffusion coefficient in the air of 20 mm^2^ s^−1^. Airflow and contaminants information are then used to determine the ^222^Rn and CO_2_ concentrations within the zone.


### 2.7. Ventilation Scenarios

The simulation was performed for 6 different sets of ventilation scenarios, where the DVR was changed according to the legal requirements and recommendations (Table 2). Scenarios 1, 2, 3 and 4 are based on the requirements of the rules relating to the ventilation and air conditioning of buildings [19]. Scenarios 5-I, 5-II, 5-III, 5-IV are based on the recommendations of the standard SIST EN 16798-1: 2019 [25], where all four categories of indoor environment quality (I–IV) were considered and applied to residential buildings. Scenario 6 is based on the Proposal of Rules for efficient use [22] and the Proposal of TSG-1-004: 2021 [23].

The calculated concentrations of ^222^Rn and CO_2_ for all of the variants were compared with the legal requirements and recommendations presented in Table 3.

So far, the Federation of European Heating, Ventilation and Air Conditioning Associations (REHVA) has also prepared the ventilation guidelines to prevent the spread of SARS-CoV-2 in workplaces [57]; the guidelines for residential buildings have not yet been prepared.

## 3. Results

### 3.1. Results of Measured ^222^Rn and CO_2_ Concentrations and Meteorological Parameters

The results of the measurements are presented in Figure 2 for the entire period, 3–15 October 2021. Figure 2a shows the outdoor radon concentration (*C*_Rn-out_) and air temperature (*T*_out_); Figure 2b shows the indoor radon concentration (*C*_Rn-in_) and the temperature difference between the indoor and outdoor air (∆*T*, *T*_in_ − *T*_out_); and Figure 2c shows the indoor carbon dioxide concentration ($C_{{\mathrm{CO}}_{2}}$).

Outdoor ^222^Rn concentrations (Figure 2a) range from 3.3 to 39 Bq m^−3^ with the average and standard deviation of 13.7 ± 7.0 Bq m^−3^. A typical daily run, with the highest concentrations in the early morning and the lowest in the afternoon, is not always pronounced. The outdoor temperature (range 4.1–24.4 °C and average 12.5 ± 4.1 °C) decreases rapidly from the beginning to the end of the measurement period. It rarely drops below 14 °C in the first days, hovers around 14 °C in the next two days (7–8 October), and is mostly below 14 °C in the last days (9–15 October). The correlation of *C*_Rn-out_ with *T*_out_ is weakly negative (*r* = 0.34), and of *C*_Rn-out_ with the pressure time gradient (∆*P*/∆*t*) in the hourly scale is very weakly negative (*r* = 0.09). A high correlation was not expected because outdoor ^222^Rn concentration is a sum of local (exhalation from the ground) and synoptic (remote) sources [15]. The contribution of each source was not sought because, in this study, only the outdoor ^222^Rn concentration during the ventilation of the apartment was needed for the simulations. Figure 2a does not show the relative air humidity (range 46–94% and average 78 ± 12%).

Due to a relatively low indoor ^222^Rn concentration (range 5–151 Bq m^−3^ and average 57 ± 30 Bq m^−3^) and the lower sensitivity of the instrument (the average error of a single measurement is ±32%), the hourly values fluctuate significantly (Figure 2b). A longer integration time of the measurements (e.g., 3 h) would give a smoother curve, but less information about the decrease in ^222^Rn concentration due to the ventilation of the apartment. The indoor ^222^Rn concentration and the temperature difference *T*_in_ − *T*_out_ show a weak positive correlation (*r* = 0.32) for the entire measurements.

In the first part of the measurements (3–8 October), when the door was opened to maintain the CO_2_ concentration below 1000 ppm, the ^222^Rn concentration also remained below 100 Bq m^−3^ for most of the time (range 5–149 Bq m^−3^ and average 46 ± 23 Bq m^−3^). In the second part (9–10 October, weekend), no ventilation on Saturday and the previous ventilation regime on Sunday were applied, and the following indoor ^222^Rn concentrations were obtained: Saturday (9 October) 32–141 Bq m^−3^ (80 ± 23 Bq m^−3^) and Sunday (10 October) 5–91 Bq m^−3^ (42 ± 23 Bq m^−3^). In the last part (11, 12, 14 October), when the ventilation was minimised to once or twice per day, ^222^Rn concentrations in the range of 14–123 Bq m^−3^ and an average of 66 ± Bq m^−3^ were obtained. On 13 October, the apartment was not ventilated and ^222^Rn concentration in the range of 36–151 Bq m^−3^ (93 ± 32 Bq m^−3^) was obtained, which is similar to October 9 when the apartment was also not ventilated (Figure 2b). The average ^222^Rn concentration in the non-ventilated apartment (87 Bq m^−3^) dropped by about 25% (66 Bq m^−3^) when ventilated once to twice per day, and by about 50% (45 Bq m^−3^) when ventilated more frequently.

The CO_2_ concentrations (Figure 2c) range from 400 to 2340 ppm with the average and standard deviation of 1010 ± 490 ppm for the entire period of measurements. Similar to indoor ^222^Rn concentration, indoor CO_2_ concentration also fluctuates according to the frequency of the ventilation (Figure 2c). In the first part (3–8 October), when the apartment was ventilated three to five times per day (except Friday, when it was not occupied for 9.5 h) to keep the CO_2_ concentration below 1000 ppm, the $C_{{\mathrm{CO}}_{2}}$ was 420–1490 ppm (average 759 ± 222 ppm). In the second part (9–10 October, weekend), under closed conditions on Saturday and frequent ventilation on Sunday, the following CO_2_ concentrations were observed: 1435–2220 ppm (average 1860 ± 260 ppm) on Saturday (9 October) and 410–2070 ppm (average 812 ± 338 ppm) on Sunday (October 10). In the last part (11, 12, 14 October), during minimal ventilation (once to twice per day), CO_2_ concentration in the range of 400–2045 ppm (average 1033 ± 424 ppm) was obtained. On 13 October, when no ventilation was performed, the range of 1120–2340 (average 1800 ± 311 ppm) was recorded, similar to 9 October.

The indoor air temperature (range 17.7–25.9 °C, average 22.4 ± 1.6 °C) and indoor air humidity (range 42–59%, average 50 ± 5%) are not presented in Figure 2. Although they are influenced by ventilation, they have less impact on the results as the temperature difference between indoor and outdoor air.

### 3.2. Comparison of Measured and Simulated ^222^Rn and CO_2_ Concentrations

Figure 3a shows a comparison of the measured and simulated datasets of ^222^Rn concentrations (*C*_Rn-m_, *C*_Rn-s_) and a similar trend of both curves is observed. In the first part (3–8 October), when the apartment was ventilated several times per day, a moderate correlation between the measured and simulated datasets is obtained (*r* = 0.62). In the second part (9–10 October, weekend), there was no ventilation on Saturday (exhibiting moderate correlation, *r* = 0.59) and frequent ventilation on Sunday (with slight correlation, *r* = 0.32). In the last part (11, 12, 14 October), the ventilation was done once or twice per day (moderate correlation, *r* = 0.68) and 13 October was without ventilation (moderate correlation, *r* = 0.47). The differences between *C*_Rn-m_ and *C*_Rn-s_ are as follows: 12 ± 20 Bq m^−3^ (3–8 October); 1 Bq m^−3^ ± 20 Bq m^−3^ (9 October); 21 ± 22 Bq m^−3^ (10 October); 9 ± 19 Bq m^−3^ (11 October); 14 ± 33 Bq m^−3^ (12 October); 14 ± 33 Bq m^−3^ (14 October); and 79 ± 29 Bq m^−3^ (13 October).

The measured and simulated concentrations of CO_2_ ($C_{{\mathrm{CO}}_{2}-m}{,\;C}_{{\mathrm{CO}}_{2}-s}$) are shown in Figure 3b. During the days of frequent ventilation (3–8 October) a moderate correlation is obtained (*r* = 0.55). In a closed condition with no ventilation (9 October), a very high correlation (*r* = 0.94) is observed, and in well ventilated (seven times) conditions (10 October) a moderate correlation (*r* = 0.69) is observed. In a poor ventilated condition (11, 12, 14 October, a moderate correlation (*r* = 0.55) is observed, and in a no ventilated condition (13 October), very weak negative correlation (*r* = −0.06) is observed. The difference between $C_{{\mathrm{CO}}_{2}-m}$ and $C_{{\mathrm{CO}}_{2}-s}$ is as follows: 151 ± 110 ppm (3–8 October, without night time, when the highest discrepancy was observed); 252 ± 164 ppm (9 October); 107 ± 134 ppm (10 October); 107 ± 93 ppm (11 October); 51 ± 181 ppm (12 October); 837 ± 278 ppm (14 October); and 281 ± 375 ppm (13 October). The exact ventilation data referred to in the above data are summarised in Table 2.

### 3.3. The Influence of Required and Recommended DVRs on Simulated ^222^Rn and CO_2_ Concentrations

The simulated concentrations are shown in Figure 4a for ^222^Rn (*C*_Rn-s_), and in Figure 4b for CO_2_ ($C_{{\mathrm{CO}}_{2}-s}$) by varying the DVRs in the apartment (Table 3) within six sets of scenarios. Deviations (expressed in h and % of simulated time, 288 h in total) from the limit values of 100 Bq m^−3^ for ^222^Rn concentration [25,59], and 1000 ppm [60] and 800 ppm for CO_2_ concentration [17], are presented in Table 4.

The best scenario is represented by the DVR in case 5C_Cat I (46.9 m^3^ h^−1^ (0.7 ACH)), recommended by EN 16798-1 [25]. For this case, the simulated ^222^Rn and CO_2_ concentrations were below the limit values (100 Bq m^−3^, 1000 and 800 ppm) for the entire simulation (the total 288 h). The worst scenarios are represented by the DVRs in the 1st and 2nd case (13.9 m^3^ h^−1^ and 15.0 m^3^ h^−1^ (0.2 ACH)), required by the rules relating to the ventilation and air conditioning of buildings [19]. In the case of 13.9 m^3^ h^−1^ (1st case), the simulated CO_2_ concentrations exceeded 1000 ppm 64% of the time (185 h), and 800 ppm 82% of the time (237 h); the simulated ^222^Rn concentration exceeded 100 Bq m^−3^ 4% of the time (10 h). In the case of 15.0 m^3^ h^−1^ (0.2 ACH), simulated CO_2_ concentration exceeded 1000 ppm 61% of the time (176 h), and 800 ppm 93% of the time (267 h); the simulated Rn concentrations exceeded 100 Bq m^−3^ 2% of the time (6 h). A similar deviation was also found for cases 5A_Cat III and 5B_Cat III, both representing categories III of indoor environmental quality (IEQ) defined by EN 16798-1 [25]. The DVRs in the scenarios (i.e., 3, 6, 5A_ Cat I, 5B_Cat I, 5C_Cat II) resulted in a simulated ^222^Rn concentration below the limit values of 100 Bq m^−3^ and a CO_2_ concentration below 1000 ppm, but not below 800 ppm.

In the last step of the study, the optimal DVRs were simulated. As can be seen in Figure 5a, they permanently assure a concentration of ^222^Rn below 100 Bq m^−3^, and concentrations of CO_2_ below 1000 ppm (Figure 5b), and below 800 ppm (Figure 5c).

## 4. Discussion

The deterioration of IAQ in residential buildings is a subject of numerous studies worldwide. One of the main features of energy-efficient buildings is increased airtightness, which leads to lower air leakage through the building envelope [61]. A controlled infiltration rate adjacent to overlooked building ventilation might result in elevated indoor air pollutant concentrations [62]. Therefore, it is not surprising that recent research has highlighted the dependence of indoor air pollutant concentrations on ventilation, either based on measured or modelled data or as a synthesis of both [63].

Dealing with ^222^Rn and CO_2_, the authors study the effects of ventilation on the measured and simulated concentrations separately, either on ^222^Rn or CO_2_. In our research, we evaluated the results of measurements and simulations of both pollutants simultaneously (Figure 3).

In the indoor air of the test apartment in our study, the measured average ^222^Rn concentration of 57 ± 30 Bq m^−3^ (range 5–151 Bq m^−3^) is about 4-fold higher than in outdoor air (13.7 ± 7.0 Bq m^−3^, 3.3–39 Bq m^−3^) (Figure 2a,b). The primary indoor ^222^Rn source is assumed ^222^Rn diffusion from building materials [64,65]. The apartment has one outdoor wall, two walls border the staircase, and one adjoins the next apartment. A minor source of ^222^Rn could be attributed to infiltration through the walls from the apartment next to and below, and less to infiltration from the staircase, where the window is open most of the time. Similar to our results, the average measured ^222^Rn concentrations obtained by García-Tobar [45] were: 62 ± 17 Bq m^−3^ (dwelling A) and 77 ± 20 Bq m^−3^ (dwelling B), with the highest values in the hallway and bathroom (115 Bq m^−3^ in dwelling A, and 150 Bq m^−3^ in dwelling B), mainly due to lower ventilation rates. Similar ^222^Rn concentrations were also obtained in the following study (62 ± 5 Bq m^−3^ in dwelling A, 78 ± 6 Bq m^−3^ in dwelling B) in which García-Tobar [46] analysed the weather factors on indoor ^222^Rn concentration. The most significant impacts on indoor Rn concentrations are reported to be wind speed and wind direction, followed by air temperature and barometric pressure. On the contrary, in our study, the dominant factor that affected ^222^Rn and CO_2_ concentrations in the non-ventilation period was the difference between the indoor and outdoor air temperature. The other parameters, such as the wind speed and direction, were not analysed.

In our study, the indoor CO_2_ concentration reflects the resident presence. At the time of the experiment, only one person was present, according to the records in Table 2. Considering that during the 12-day measurements (Figure 2c), the apartment was intensively ventilated for 6 days, poorly ventilated for 4 days and not ventilated for 2 days, the average concentration of 1010 ± 490 is relatively high (range 400–2340 ppm). Similar findings have been reported for naturally [40,41,42,43] and mechanically ventilated buildings [43,44]. A cross-sectional study in 79 Greenlandic dwellings in the town of Sisimiut found that 73% of bedrooms were insufficiently ventilated ($C_{{\mathrm{CO}}_{2}}$ > 1000 ppm), that younger dwellings (built after 1990) had poorer IAQ than older dwellings, and that children’s bedrooms (2000–4000 ppm) had higher CO_2_ concentrations than adults’ bedrooms [40].

Our results showed that the actual ventilation rate (controlled by ventilation; uncontrolled by infiltration, breakthroughs, shafts) significantly influences the accuracy of the simulation. When the accurate occupancy schedule and ventilation rates were included in the model, the measured and simulated data of ^222^Rn and CO_2_ concentrations were well-matched (Figure 3).

A comparison of the measured and simulated time series of ^222^Rn concentrations (Figure 3a) shows a moderate correlation of frequent ventilation days (*r* = 0.62), sparse ventilation days (*r* = 0.68), and no ventilation days (*r* = 0.59 and *r* = 0.47). The slight correlation (*r* = 0.32) on 10 October, when the maximum number of ventilations was performed, revealed a too low ^222^Rn monitor sensitivity for this concentration range, which resulted in the unreliable ^222^Rn generation rates used in the simulation. In general, the time series of simulated ^222^Rn concentration is underestimated, except during the last 3 days (12–14 October), when it is significantly overestimated. In the last 3 days, the outside air temperature dropped sharply overnight (Figure 2a), leading to a temperature difference ∆*T* between the outdoor and indoor air (Figure 2b) sufficiently high to trigger the so-called ‘chimney effect’, a natural draft of air through the chimney, and thus decreased the indoor ^222^Rn concentration. This type of (uncontrolled) ventilation is not considered in our model and, therefore, leads to the overestimation in the simulation in this period (Figure 3a). In the study by García-Tobar [45], ^222^Rn data are compared for: (i) average measured and simulated values (62/66 Bq m^−3^ in dwelling A, and 77/74 Bq m^−3^ in dwelling B); and (ii) the highest measured and simulated values (115/110 Bq m^−3^ in dwelling A, and 150/110 Bq m^−3^ in dwelling B). When the flow rate of the fan in the bathroom was doubled (from 2 to 4 dm^3^/(s m^−2^)), the ^222^Rn concentration was reduced by 50% (from 110 to 55 Bq m^−3^) and the following measured and simulated average ^222^Rn concentrations obtained: 62 ± 5 Bq m^−3^/75 ± 3 Bq m^−3^ (dwelling A) and 78 ± 6 Bq m^−3^/83 ± 4 Bq m^−3^ (dwelling B), with the Pearson correlation coefficients of 0.341 (dwelling A) and 0.198 (dwelling B).

The measured and simulated concentrations of CO_2_ in our study (Figure 3b) show moderate correlation with the same correlation coefficient (*r* = 0.55) during the days of frequent ventilation and poor ventilation. A very high correlation (*r* = 0.94) was obtained on 9 October under no ventilation, and the resident was present all day. On the other hand, under a no ventilation condition, on 13 October, a very weak negative correlation (*r* = −0.06) was noticed, most probably due to the chimney effect not considered in our simulation (uncontrolled ventilation). The curve of $C_{{\mathrm{CO}}_{2}-s}$ is underestimated during the whole dataset, and the discrepancy is more pronounced in the last part. The reason for the sudden overestimation on the last day has not been understood yet. A similar study carried out by Szczepanik-Ścisło and Flaga-Maryańczyk [44] also performed a CONTAM 3.2 simulation, where the influence of occupancy schedules and the ventilation efficiency on the CO_2_ concentration was analysed over 10 days. In the first case, when the simulations were conducted at the minimum ventilation rate and the door to the wardrobe was open, the simulation data were mainly higher than the measurement (300–800 ppm). In the second case, with the minimum ventilation rate and a closed door, the difference between the measurement and the simulation data was 600–1000 ppm. In the following case, with medium ventilation and both doors open, the measurement data were higher than those of the simulation (100–600 ppm) in the first and the last part of the period, and lower (approximately 400 ppm) in the middle part of the period. In the last case, which included the occupancy schedule and the real ventilation rate, the simulated values were 100–300 ppm lower than the measured ones. Regarding the results, the authors proved that the CONTAM 3.2 program was able to recreate the conditions of CO_2_ inside the analysed room, especially if the occupancy schedules and real ventilation rate (with all openings) were precisely considered in the simulation.

Studies that simultaneously address ^222^Rn and CO_2_ are rare, especially on the critical analysis of the DVR. Our study analysed the influence of required and recommended DVRs on CO_2_ and ^222^Rn concentrations with six sets of scenarios. The best scenario is the DVR in case 5C_Cat I (46.9 m^3^ h^−1^ (0.7 ACH)), recommended by EN 16798-1 [25], which resulted in the lowest CO_2_ (656 ± 121 ppm) and ^222^Rn concentrations (57 ± 13 Bq m^−3^). The worst scenarios are the DVRs in the 1st and 2nd case (13.9 m^3^ h^−1^ and 15.0 m^3^ h^−1^ (0.2 ACH)), required by the rules relating to the ventilation and air conditioning of buildings [19], as well as the DVRs in cases 5A_Cat III and 5B_Cat III (15.0 m^3^ h^−1^; 14.4 m^3^ h^−1^ (0.2 ACH)), defined by EN 16798-1 [25]. For example, scenario two with 15 m^3^ h^−1^ resulted in CO_2_ (1159 ± 291 ppm) and ^222^Rn concentration (59 ± 21 Bq m^−3^), and scenario 5B_Cat III with 14.4 m^3^ h^−1^ (0.2 ACH) resulted in CO_2_ (1188 ± 300 ppm) and ^222^Rn concentration (61 ± 21 Bq m^−3^). This was the case for the scenarios defined for the III and IV categories of IEQ. 5A_Cat III and 5B_Cat III resulted in concentrations exceeding the limits for *C*_Rn-s_ (6 h above and 8 h above 100 Bq m^−3^) and $C_{{\mathrm{CO}}_{2}-s}$ (1159 ± 291 ppm and 1188 ± 300 ppm). Moreover, category III of IEQ is also defined by the national Proposal of Rules on efficient use [22] and Proposal of TSG-1-004:2021 [23].

Particularly in residential buildings with a lower net floor area per occupant, and therefore, lower DVRs (if defined per floor area), higher CO_2_ emissions result. Several authors have pointed out the same problem. Bekö et al. [42] inspected Danish homes and found that in 57% of new dwellings, the ventilation rate is lower than the minimum required 0.5 h^−1^, in 32% of bedrooms, an average $C_{{\mathrm{CO}}_{2}}$ < 1000 ppm during the measured nights; in 23% of rooms, at least 20 min during the night a $C_{{\mathrm{CO}}_{2}}$ > 2000 ppm; and in 6% of rooms a $C_{{\mathrm{CO}}_{2}}$ > 3000 ppm, which is similar to the findings of Kotol et al. [40]. The lower ventilation rate problem was also addressed in a detached passive house in the Silesian region of Poland [44]. The stated reason for the increased CO_2_ concentration in the bedroom (peak 1800 ppm) was the minimised ventilation rate to reduce the noise of the ventilation system. Additionally, Sekkhar and Goh [43] conducted a study in naturally and mechanically ventilated and air-conditioned bedrooms in Singapore’s hot and humid climate. Higher CO_2_ concentrations in their study were related to the use of split-system air-conditioning units that only recirculate air and do not provide fresh outdoor air. Increased DVRs, however, resulted not only in the reduction in CO_2_ but also in the reduction in ^222^Rn concentrations. In the García-Tobar [45] study, doubling the fan’s flow rate in the bathroom reduced the ^222^Rn concentration by 50%. The significant reduction in ^222^Rn concentration due to increasing the DVR was also the case in our study.

This problem needs to be understood in the broader context of housing policy. For example, in Slovenia, according to the Statistical Office of the Republic of Slovenia [66] for 2018, 38% of housing consisted of small apartments with a net floor area of less than 50 m^2^ (i.e., studio, one- and two-bedroom apartments). Two-thirds of the population of Slovenia reside in one- or two-bedroom apartment buildings, and, therefore, the percentage of people living in underoccupied apartments is 30.4% (in 2018). Compared to the EU-27, the percentage is slightly higher, at 33.0%, and the most exposed populations are young people and children [67]. Smaller apartments are, thus, the most critical in terms of CO_2_, particularly in the presence of overcrowding. However, following the Energy Performance of Buildings Directive [29,30], mechanical ventilation with heat recovery has become an unavoidable element of a nearly zero energy building (NZEB); and according to the current construction practice [26], they often operate on too low DVRs. Consequentially, poor IAQ might be in contradiction to the objectives of the Resolution on the National Housing Programme 2015–2025 [68] for quality and functional housing. ^222^Rn can further impair IAQ, especially in low-floor apartments directly connected to the ground. On upper floors, ^222^Rn usually does not pose any significant health threat. However, more frequent ventilation, which lowers the CO_2_ concentration, also lowers the ^222^Rn concentration and thus, further reduces the health risk.

Based on the results of our research, optimal DVRs are proposed that can be regulated by demand-controlled ventilation using sensory information [69] or as an automatic control system [26], which may also include other health risks of chemical or biological origin [70]. As simulated in our study (Figure 5), to ensure CO_2_ below 1000 ppm and ^222^Rn below 100 Bq m^−3^, permanent ventilation of at least 36.6 m^3^ h^−1^ (0.5 ACH) is required. To ensure CO_2_ below 800 ppm, the DVR must always be at least 46.9 m^3^ h^−1^ (0.7 ACH).

The simultaneous analysis of measured and simulated ^222^Rn and CO_2_ data highlights the benefits in data evaluations compared to studies of either ^222^Rn or CO_2_, as in our previous work [10,11,26]. However, the most beneficial evaluation was the transient analysis on measured and simulated ^222^Rn and CO_2_ datasets, which we used for the first time. There are some further steps foreseen in our future research, which might improve the transient analysis of the datasets: (i) to add meteorological parameters, e.g., wind speed and direction, precipitation, etc.; (ii) to provide detailed information on the infiltration rate (which was considered in a simplified manner and defined as a constant value, as the primary purpose was to examine the quantitative requirements for DVRs with the controlled ventilation); and (iii) to prolong the measurement duration to all yearly seasons (in the present study, simulations were based on a short period, in which natural ventilation was executed to examine only the characteristic of high, moderate and no ventilation periods). In particular, our future work will upgrade the methodology for characterizing the IAQ with measurements and simulations in other buildings types, such as family houses and non-residential buildings. The effectiveness of natural and mechanical ventilation will also be evaluated, and the optimal DVRs examined.

## 5. Conclusions

Based on the measurements, ^222^Rn and CO_2_ concentrations in a small apartment were simulated by using the CONTAM 3.4 program and six sets of the DVR scenarios modelled. The optimal DVRs, which permanently assure ^222^Rn concentrations below 100 Bq m^−3^, and CO_2_ concentrations below 1000 ppm and 800 ppm, were sought. The main findings of the research are as follows:A comparison of measured and simulated time series of ^222^Rn and CO_2_ concentrations shows a moderate correlation (*r* = 0.62 for ^222^Rn and 0.55 for CO_2_) during the days of frequent ventilation, which was our main focus in the study.A critical analysis of six sets of ventilation scenarios showed that the optimal DVR values were those defined as the maximum amounts of fresh air, determined per floor area and per person, and applied for category I of indoor environmental quality (for test apartment: 5C_Cat I with 46.9 m^3^ h^−1^ (0.7 ACH) that resulted in 656 ± 121 ppm, 57 ± 13 Bq m^−3^). Lower DVR values, especially those defined for categories III or IV of IEQ (5A_Cat III with 15.0 m^3^ h^−1^ and 5B_Cat III with 14.4 m^3^ h^−1^ (0.2 ACH)), resulted in CO_2_ and ^222^Rn concentrations above the limit values (CO_2_: 1159 ± 291 ppm and 1188 ± 300 ppm; ^222^Rn: 6 h above and 8 h above 100 Bq m^−3^), which can present a problem for buildings located in Zone 3 areas.To increase the accuracy of our simulation, a more extended time series of measured data is needed, which should include all seasons of the year.Although the measured and simulated data matched relatively well, uncontrolled air infiltration through the building envelope needs to be further studied and defined to improve the model.The approach presented in our study can be applied to various building types to determine the optimal DVR values for ventilation. However, special attention should be paid to small apartments which, in the EU, have a high overcrowding rate of 33.0% (in Slovenia 30.4%). Accordingly, to protect sensitive and fragile occupants, a sufficient amount of fresh air volume for category I of indoor environmental quality has to be provided in terms of CO_2_. In addition, by lowering the CO_2_ concentration, the ^222^Rn concentration is also reduced, thus minimising the health risk.Our findings might be implemented in national legislation and the existing construction practice, which will result in safer and healthier indoor environments.

## Figures and Tables

**Figure 1 ijerph-19-02125-f001:**
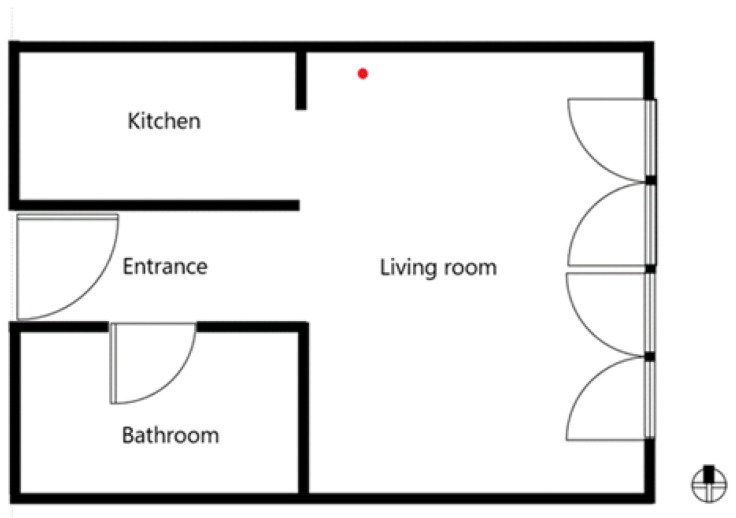
Floor plan of the tested apartment ventilation zone (red dot indicates the location of the instruments).

**Figure 2 ijerph-19-02125-f002:**
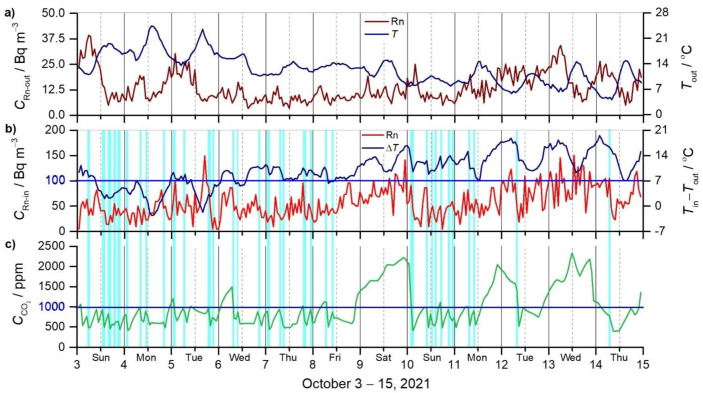
Results of the measurements in the period 3–15 October 2021: (**a**) radon concentration (*C*_Rn-out_) and air temperature (*T*_out_) outdoors; (**b**) the radon concentration (*C*_Rn-in_) indoors and the temperature difference between the indoor and outdoor air (∆*T* = *T*_in_ − *T*_out_); (**c**) the carbon dioxide concentration ($C_{{\mathrm{CO}}_{2}}$) indoors. The blue regions in (**b**,**c**) indicate the ventilation periods, and the blue lines in (**b**) the Rn limit according to WHO recommendations [59] and the CO_2_ limit according to [60], respectively. The solid lines indicate midnight, and the broken lines indicate noon in the gridlines.

**Figure 3 ijerph-19-02125-f003:**
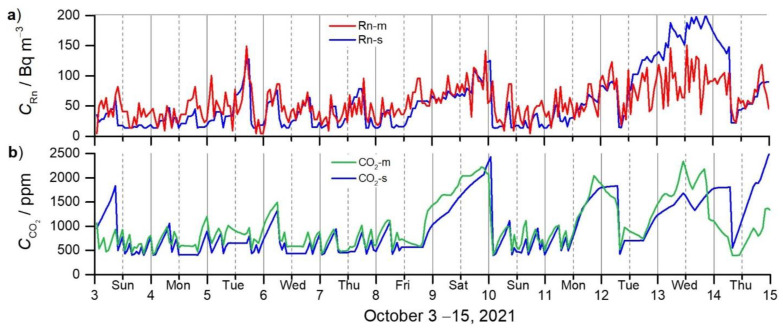
Measured and simulated concentrations of: (**a**) radon *C*_Rn-m_ and *C*_Rn-s_ [Bq m^−3^]; and (**b**) carbon dioxide $C_{{\mathrm{CO}}_{2}-m}$ and $C_{{\mathrm{CO}}_{2}-s}$ [ppm] in the period 3–15 October 2021. The solid lines indicate midnight and the broken lines indicate noon in the gridlines.

**Figure 4 ijerph-19-02125-f004:**
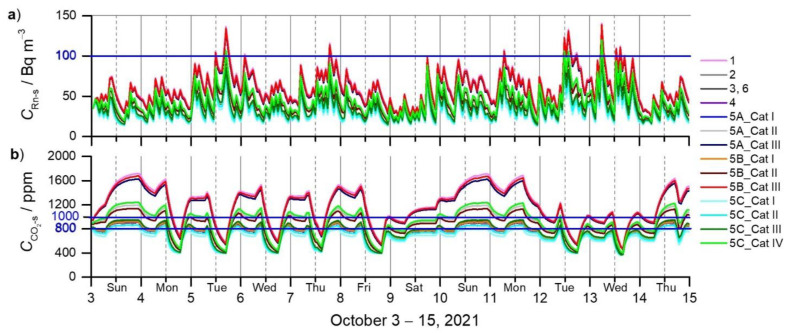
Simulated concentrations of: (**a**) radon, *C*_Rn-s_ [Bq m^−3^]; and (**b**) carbon dioxide ($C_{{\mathrm{CO}}_{2}-s}$) [ppm] in the apartment for 6 sets of scenarios in the period October 3–15, 2021. The blue lines indicate the Rn limit according to WHO recommendations [59] and the CO_2_ limits according to [60] and [17], respectively. The solid lines indicate midnight and the broken lines indicate noon in the gridlines.

**Figure 5 ijerph-19-02125-f005:**
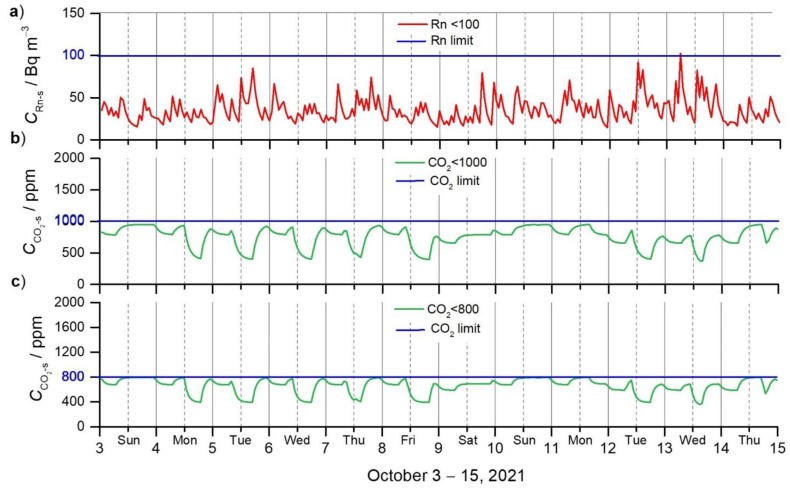
Optimal DVR in the apartment (3–15 October 2021), based on the criteria: (**a**) *C*_Rn-s_ < 100 Bq m^−3^; (**b**) $C_{{\mathrm{CO}}_{2}-s}$ < 1000 ppm; and (**c**) $C_{{\mathrm{CO}}_{2}-s}$ < 800 ppm. In the gridlines, the solid line indicates midnight and the broken line is noon.

**Table 1 ijerph-19-02125-t001:** Ventilation schedule of the apartment (except in the periods of absence, one person was present).

Day of the Week, Date	Absence Time Start–End	Absence Duration [h]	Ventilation Time Start–End	Ventilation Duration [h]
Sunday, 03.10.2021			09:40–10:00 12:40–13:20 15:30–16:50 17:55–18:50 20:00–21:15	0.33 0.67 1.33 0.92 1.25
Monday, 04.10.2021	12:00–19:00	7.00	00:30–01:50 08:00–08:35 11:05–11:55 19:30–20:25	1.33 0.58 0.83 0.92
Tuesday, 05.10.2021	8:30–17:00	8.50	00:58–01:33 06:20–06:45 18:40–19:30 21:00–21:40	0.58 0.42 0.83 0.67
Wednesday, 06.10.2021	10:00–18:15	8.25	06:50–07:20 09:35–09:55 20:05–20:50	0.50 0.33 0.75
Thursday, 07.10.2021	8:50–12:30 12:40–15:15	3.67 2.58	00:35–01:35 07:15–07:41 08:35–08:46 18:55–19:50 22:50–23:20	1.00 0.43 0.18 0.92 0.50
Friday, 08.10.2021	10:30–20:00	9.50	06:35–07:10 09:51–10:00	0.58 0.15
Saturday, 09.10.2021	–	–	–	–
Sunday, 10.10.2021			01:03–02:42 09:30–10:00 11:50–12:30 13:00–13:40 16:32–17:05 21:06–21:26 23:28–23:55	1.65 0.50 0.67 0.67 0.55 0.33 0.45
Monday, 11.10.2021			06:40–06:55	0.25
			09:30–09:48	0.30
Tuesday, 12.10.2021	10:00–18:20	8.33	07:30–08:00	0.50
Wednesday, 13.10.2021	11:15–15:55	4.67	–	–
Thursday, 14.10.2021	17:30–19:25	1.92	07:15–07:35	0.33

**Table 2 ijerph-19-02125-t002:** List of scenarios with the required and/or recommended design ventilation rate (DVR).

Scenario	Level of Obligation	Required, Recommended DVR			Reference
		Descriptive Criterion	Quantitative Criterion General	Quantitative Criterion Test Apartment	
1	Requirement	Minimal air changes per hour (ACH) in the absence of occupants to remove building emissions and prevent harm (can be considered in the 24 h cycle)	0.20 h^−1^	13.9 m^3^ h^−1^ (0.2 ACH)	[19]
2	Requirement	Minimal outdoor air intake	15.0 m^3^ h^−1^ person^−1^	15.0 m^3^ h^−1^ (0.2 ACH)	[19]
3=6	Requirement	Minimal ACH	0.50 h^−1^	34.6 m^3^ h^−1^ (0.5 ACH)	[19,22,23]
4	Requirement	Minimal volume of air per floor surface area (without consideration of other sources)	1.50 m^3^ h^−1^ m^−2^	40.0 m^3^ h^−1^ (0.6 ACH)	[19]
5A: Cat I-III	Recommendation	Ventilation rate per person and per m^2^ floor area	Cat I: 12.6 m^3^ h^−1^ person^−1^ + 0.9 m^3^ h^−1^ m^−2^ Cat II: 9.0 m^3^ h^−1^ person^−1^ + 0.54 m^3^ h^−1^ m^−2^ Cat III: 5.4 m^3^ h^−1^ person^−1^ + 0.36 m^3^ h^−1^ m^−2^	36.6 m^3^ h^−1^ (0.5 ACH) 23.4 m^3^ h^−1^ (0.3 ACH) 15.0 m^3^ h^−1^ (0.2 ACH)	[25]
5B: Cat I-III	Recommendation	Ventilation rate per person	Cat I: 36.0 m^3^ h^−1^ person^−1^ Cat II: 25.2 m^3^ h^−1^ person^−1^ Cat III: 14.4 m^3^ h^−1^ person^−1^	36.0 m^3^ h^−1^ (0.5 ACH) 25.2 m^3^ h^−1^ (0.4 ACH) 14.4 m^3^ h^−1^ (0.2 ACH)	[25]
5C: Cat I-IV	Recommendation	Ventilation rate per m^2^ floor area with infiltration	Cat I: 1.76 m^3^ h^−1^ m^−2^ Cat II: 1.51 m^3^ h^−1^ m^−2^ Cat III: 1.26 m^3^ h^−1^ m^−2^ Cat IV: 0.83 m^3^ h^−1^ m^−2^	46.9 m^3^ h^−1^ (0.7 ACH) 40.2 m^3^ h^−1^ (0.6 ACH) 33.6 m^3^ h^−1^ (0.5 ACH) 22.1 m^3^ h^−1^ (0.3 ACH)	[25]

**Table 3 ijerph-19-02125-t003:** Requirements and recommendations for the concentrations in indoor air of: (a) radon (^222^Rn) [19,25,28,58,59]; and (b) carbon dioxide (CO_2_) [17,19,60].

Obligatory Level	Required, Recommended Concentration	Reference
**(a) ^222^Rn**		
Requirement: the permissible value of Rn in indoor air	400 Bq m^−3^	[19]
Requirement: the reference level of the average annual concentration of radon in closed living and working spaces	300 Bq m^−3^	[28]
Recommendation: WHO guideline value	100 Bq m^−3^	[25,59]
Recommendation: WELL Building Standard. The following conditions are met in projects with regularly occupied spaces at or below grade: radon less than 4 pCi/L in the lowest occupied level	4 pCi L^−1^ (148 Bq m^−3^)	[58]
**(b) CO_2_**		
Requirement: the permissible value of CO_2_ in indoor air	1667 ppm	[19]
Recommendation: for the design and assessment of energy performance in buildings	Cat I: 350 ppm ^a^ Cat II: 500 ppm ^a^ Cat III: 800 ppm ^a^ Cat IV: <800 ppm ^a^	[17]
Recommendation:	Max: 2500 ppm Recommended: 1000 ppm	[60]

Note: ^a^ value above outdoor background concentration. Cat I: presents a high level of expectation and is recommended for spaces occupied by very sensitive and fragile persons with special requirements such as disabled, sick, very young children, and elderly persons; Cat II: normal level of expectation, should be used for new buildings and renovations; Cat III: acceptable, moderate level of expectation, may be used for existing buildings; Cat IV: values outside the criteria for the above categories. The last category should only be accepted for a limited part of the year [17].

**Table 4 ijerph-19-02125-t004:** Deviations of simulated *C*_Rn-s_ [Bq m^−3^] and $C_{{\mathrm{CO}}_{2}-s}$ [ppm] from the limit values [17,25,59,60] for 6 sets of scenarios.

Scenario	DVR	Duration of $C_{{\mathrm{CO}}_{2}-s}$ above 1000 ppm		Duration of $C_{{\mathrm{CO}}_{2}-s}$ above 800 ppm		Duration of *C*_Rn-s_ above 100 Bq m^−3^	
		[h]	[%]	[h]	[%]	[h]	[%]
1	13.9 m^3^ h^−1^ (0.2 ACH)	185	64	237	82	10	4
2	15.0 m^3^ h^−1^ (0.2 ACH)	176	61	267	93	6	2
3=6	34.6 m^3^ h^−1^ (0.5 ACH)	0	0	93	32	0	0
4	40.0 m^3^ h^−1^ (0.6 ACH)	0	0	60	21	0	0
5A_ Cat I-III	36.6 m^3^ h^−1^ (0.5 ACH)	0	0	83	29	0	0
	23.4 m^3^ h^−1^ (0.3 ACH)	87	30	169	59	2	1
	15.0 m^3^ h^−1^ (0.2 ACH)	176	61	218	76	6	2
5B_ Cat I-III	36.0 m^3^ h^−1^ (0.5 ACH)	0	0	81	28	0	0
	25.2 m^3^ h^−1^ (0.4 ACH)	61	21	159	55	1	0.4
	14.4 m^3^ h^−1^ (0.2 ACH)	188	65	226	79	8	3
5C_ Cat I-IV	46.9 m^3^ h^−1^ (0.7 ACH)	0	0	0	0	0	0
	40.2 m^3^ h^−1^ (0.6 ACH)	0	0	61	21	0	0
	33.6 m^3^ h^−1^ (0.5 ACH)	0	0	133	46	0.5	0.2
	22.1 m^3^ h^−1^ (0.3 ACH)	117	41	163	57	2	0.7

## Data Availability

Not applicable.

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
