# Peer review of "Analysis of Ventilation Efficiency as Simultaneous Control of Radon and Carbon Dioxide Levels in Indoor Air Applying Transient Modelling"

_ijerph, 2022, doi:10.3390/ijerph19042125_

Round 1
Reviewer 1 Report
The manuscript ‘Analysis of ventilation efficiency as simultaneous control of radon and carbon dioxide levels in indoor air applying transient modelling’ by Dovjak, M. et al. presents a well-structured study comparing physical measurements of CO2 and Rn to those simulated in CONTAM in a residential setting. The Literature review, rather than discussing studies in the area, seems to be defining the different terms used in this paper rather than discussing different studies on which this paper focuses ‘ventilation efficiency of residential buildings.’ How do the authors find the research gap for studying this topic? The authors claim to have conducted measurements in two sites, but only one of them is properly described. Further, the results & conclusions only focus on the results of one site. Some detailed comments are below:
L32 - Does the built environment need to be in bold?
L33-34 - Please briefly explain how these deaths are related to the built environment (i.e. IAQ, thermal comfort?)
L37 - Does the building need to be in bold?
L45, 46, 49, 54, 73, 85, 101, 110 - Do the terms need to be in bold?
L73-74 - Sentence needs a reference.
L124 - Section on Study design needs to be adequately addressed; several bullet points is not enough to define a complete section.
L138-9 - Accordingly to whom? It would be better to make reference to the Köppen climate classification and describe the weather.
L144 - If the paper centres on ‘ventilation efficiency of residential buildings,’ why was an office building selected. This needs to be carefully explained.
L149-61 - If there are two buildings selected, why one is described?
L172-77 - Please describe the airflow rates and/or CO2 achieved for each ventilation scenario; please make a reference to Table 1. This is of particular importance if carrying modelling.
L179 - Here, the authors describe that the measurements were conducted on both sites, but information about the ventilation schedules & zones are only provided for one site.
L 186 - Was there a dining table as well on the second location (office building)
L178-97 - Did the authors follow an international standard for monitoring? If so, which one, what are its characteristics? If not, why?
L198 - Table not referred to within the text up to this point.
L211-19 - How do these airflows compare to the site(s)?
L259 - What does the Colum 3 represents in Table 2
L259 & 261 - tables haven’t been cited on the text up to this point
L277 - Please add correct legends for all graphs. Please avoid the use of CCO2/CRn-in to describe concentrations of… this is confusing for the reader. They are usually described just as CO2, Rnin (address this issue through the complete manuscript).
L331-32 - Why? Do they have any impact on the results? If so, how?
L 361 - Was the model calibrated with on-site measurements? If so, show and describe it. If not, why not and why the simulated values are accurate?
L429-67 - The authors describe several studies that look at similar issues of this paper, which should have been provided in the LR. In the discussion, it would be more appropriate to discuss the content of the author’s study and compare them to other studies.
L5418 - Please describe the limitations of the study and further work.
L551 - The authors previously mentioned they did 2 sites; here they claim to have done only one?
L551-61 - Please avoid summarising the study in the conclusion section. The authors should focus on the main findings and what do they mean in the context of the study.
L602-3 - Surely, if the authors made measurements in a residential setting, they should have gained consent and ethics approval to do this.
Author Response
RESPONSE TO REVIEWERS' COMMENTS
Title: Analysis of ventilation efficiency as simultaneous control of radon and carbon dioxide levels in indoor air applying transient modelling (ijerph-1511960), International Journal of Environmental Research and Public Health. According to reviewers` comments, the authors are now submitting the corrections of the manuscript.
All the comments were very beneficial and helped us to increase the quality of the manuscript. The list of completions, written according to the reviewers' comments, is presented in the attachment (please see the attachment) and included in the manuscript's revised version (separate file). Reviewers' comments are written in italics in brackets.

Reviewer 2 Report
In this manuscript, the authors illustrate the impact of ventilation on the accumulation of Radon and CO2 indoors by measuring these fluctuations indoors and outdoors and modeling the expected results. Although the results have particular value in monitoring indoor air quality, authors should make several changes before accepting the manuscript.
The authors must justify in the introduction the interest of comparing the real indoor air data with a model of accumulation of Radon and CO2 indoors as a function of the ventilation rate.
In the Methods section, the location of the outdoor sampling should be clearly stated.
The use of different instruments for monitoring radon outdoors and indoors should be explained
I am not convinced by the organization of the discussion section. Authors' conclusions should be systematically compared to published data to capture the message and put them into perspective. It is also important to explain the interest of presenting simultaneously the results of real measurements and simulations for these two pollutants. What this fact brings new to the field.
Author Response

(The authors gave the same response as above.)

Reviewer 3 Report
The article discusses the effect of ventilation on radon and CO2 concentration build-up in an indoor built environment. The effect of ventilation rate on these gases is well known, but the novelty of the study is that it is trying to predict the concentration based on the ventilation rate. In this context,, I have the following comments:
- The CO2 measurement technique could be presented in some detail. Right now no details are presented.
- I am curious to understand how this technique could work in High Radiation Background Regions or if the building materials are made of materials with natural radioactivity higher than the normal background region? How do you account for the source term of Radon since it is an important parameter governing its concentration in the indoor environment?. This may be discussed in some detail.
Author Response
RESPONSE TO REVIEWERS' COMMENTS
Title: Analysis of ventilation efficiency as simultaneous control of radon and carbon dioxide levels in indoor air applying transient modelling (ijerph-1511960), International Journal of Environmental Research and Public Health. According to editor`s and reviewers` comments, the authors are now submitting the corrections of the manuscript.
All the comments were very beneficial and helped us to increase the quality of the manuscript. The list of completions, written according to the comments, is presented below and included in the manuscript's revised version (separate file). Reviewer's comments are written in italics in brackets.
Reviewer's comments
Reviewer 3 (round 1)
1.“The article discusses the effect of ventilation on radon and CO2 concentration build-up in an indoor built environment. The effect of ventilation rate on these gases is well known, but the novelty of the study is that it is trying to predict the concentration based on the ventilation rate. In this context, I have the following comments:
2.The CO2 measurement technique could be presented in some detail. Right now no details are presented. I am curious to understand how this technique could work in High Radiation Background Regions or if the building materials are made of materials with natural radioactivity higher than the normal background region? How do you account for the source term of Radon since it is an important parameter governing its concentration in the indoor environment?. This may be discussed in some detail«.
Dear reviewer,
We are glad that you recognised the importance of the topic. Thank you for your valuable comments and recommendations, which we answered below. According to your suggestions, we added some more details to the manuscript. We hope that the paper has been improved substantially and fulfilled your requirements.
Comments and corrections
1.“The CO2 measurement technique could be presented in some detail. Right now no details are presented.”
Thank you for your comment, we agree that in the manuscript was missing information on the CO2 measurement technique. Therefore, we corrected the methodology section in order to add the details of the measurement protocol:
P6, L209 (2. Materials and Methods; 2.5. Measurements)
The measurements were conducted in the period October 3 – 15, 2021 and all presented data are reported in local time (LST = UTC + 2h). A standardized protocol for characterizing IAQ in residential buildings was followed [16,17,19,25,50]. The instrument for continuous measurement of the selected parameters was placed in the respiratory zone (living zone) at the height of 1.1 m above the floor, 3 m from the external window and wall, door and radiator, 0.8 m from the internal wall (Figure 1) [16,17,19,25,50].
The selection of instruments was based on expected radon concentrations in indoor and outdoor air and requirements of our radon laboratory [51], accredited according to ISO / IEC 17025 [52]. Both devices were operated continuously in a diffusion mode with a frequency of once per hour.
Indoor air: radon CRn-in [Bq m–3] and carbon dioxide [ppm] concentrations, room air temperature Tin [°C] and relative air humidity RHin [%] were measured with the Radon Scout Professional device (Sarad). placed on a dining table in the living room (Figure 1). The Radon Scout Professional monitor operates in the range from 0 Bq m–3 to 2 MBq m–3 with the sensitivity to Rn >2.5 cpm / (kBq m–3). The integration interval of the data should be adjusted to the concentration range. If the expected radon concentrations are of the order of the reference level of 300 Bq m–3 or below, an interval of 60 minutes should be used. The sensor for CO2 operates in the range of 400 to 5000 ppm [53]. The integrated CO2 sensor uses the non-dispersive infrared (NDIR) operational principle.
Outdoor air: radon CRn-out [Bq m–3] concentration, temperature Tout [°C], relative humidity RHout [%], and pressure Pout [hPa] were measured with the AlphaGUARD (Bertin Instruments) monitor, placed into a Stevenson screen at a height of 1.5 m above the ground. The instrument operates in the range from 2 Bq m–3 to 2 MBq m–3, the efficiency of the detector is 1 cpm at 20 Bq m–3 [54].
2.“I am curious to understand how this technique could work in High Radiation Background Regions or if the building materials are made of materials with natural radioactivity higher than the normal background region?”
The main parameter, on which is based simulated radon, is radon generation rate (Bq/h), calculated from the measured radon concentration (entry rate during closed periods, removal rate during periods of ventilation). Therefore, we may simulate radon concentration accurately also in higher radon concentration ranges (e.g. high radiation background regions or buildings with increased radon concentration due to building materials).
However, the question arises if such radon simulations make sense when high radon concentration is measured.? When radon concentration significantly exceeds the limit value, it should be decreased by adequate mitigation measures to reduce radon concentration permanently. Ventilation is an efficient measure when we may maintain the radon concentration below the limit, not to drastically influence the living comfort of the residents and not to increase the costs for energents significantly.
In this paper, we wanted to draw attention to the problem of the high tightness of buildings nowadays, which leads to worsening indoor air quality in general. Our example of Rn and CO2 in a 3rd floor flat, where one would not expect high radon concentration, may easily exceed the WHO radon limit when poorly ventilated. Our model can optimise the frequency and duration of ventilation to keep radon and CO2 (and other pollutants) below the limit value, which, in our opinion, is the great benefit of the developed methodological approach.
- “How do you account for the source term of Radon since it is an important parameter governing its concentration in the indoor environment?. This may be discussed in some detail«.
Radon, entering the indoor space mainly through the foundation (cracks, joints) from the soil gas, reflects geology, meteorological conditions and the quality of construction (tightness of the building envelope). The radon generation (entry) rate (Bq/h) is the sum of accounts for all the above parameters, which was discussed in more detail in our study Dovjak et al. [26].
The severity of the radon problem in an area (maximum concentration in soil gas) depends on the geogenic radon potential. Its class (e.g. high, normal, low) in the area is a prerequisite for radon-related studies in the built environment (e.g. radon safe building). The building presented in our study is located in an area with low geogenic radon potential and has mechanical ventilation of the garage underneath. The radon concentration, which may reach in the non-ventilated apartment (3th floor) around 4-times the outdoor radon concentration, is assumed to enter the space from building material, as discussed in the manuscript in lines 450–467.

Round 2
Reviewer 2 Report
The authors have difficulties to present the aim of their work , to present data and discuss them by comparison to the existing literature. The modifications proposed don't answer the expectations.
Author Response
RESPONSE TO REVIEWERS' COMMENTS
Title: Analysis of ventilation efficiency as simultaneous control of radon and carbon dioxide levels in indoor air applying transient modelling (ijerph-1511960), International Journal of Environmental Research and Public Health. According to editor`s and reviewers` comments, the authors are now submitting the corrections of the manuscript.
All the comments were very beneficial and helped us to increase the quality of the manuscript. The list of completions, written according to the comments, is presented below and included in the manuscript's revised version (separate file). Reviewer's comments are written in italics in brackets.
Reviewer 2 (round 2)
- “The authors have difficulties to present the aim of their work, to present data and discuss them by comparison to the existing literature. The modifications proposed don't answer the expectations”.
Dear reviewer,
Thank you for your comment. We are sorry that our thorough modification of the manuscript, in which we consider all your comments and recommendations, did not meet your expectations.
In your first review, you pointed out some of the weak points, which we tried to correct or clarify, to increase the paper's understanding and quality, which we highly appreciate.
- To more clearly present the aim of our work, we have first summarised recent outstanding studies on the field in the Introduction and after we gave the short concept of our study (P3, 127), and in the Conclusion the primary outcome (P16, 603).
- Regarding the comment on difficulties to present data and discuss them by comparison to the existing literature, the problem of Rn and CO2, related to ventilation and high tightness of buildings, is in the broad context discussed in the Introduction and, related to our research and results, is commented in the Discussion.
